# In Vivo Imaging-Based Techniques for Early Diagnosis of Oral Potentially Malignant Disorders—Systematic Review and Meta-Analysis

**DOI:** 10.3390/ijerph182211775

**Published:** 2021-11-10

**Authors:** Marta Mazur, Artnora Ndokaj, Divyambika Catakapatri Venugopal, Michela Roberto, Cristina Albu, Maciej Jedliński, Silverio Tomao, Iole Vozza, Grzegorz Trybek, Livia Ottolenghi, Fabrizio Guerra

**Affiliations:** 1Department of Oral and Maxillo-Facial Sciences, Sapienza University, 00161 Rome, Italy; marta.mazur@uniroma1.it (M.M.); albu.1811182@studenti.uniroma1.it (C.A.); iole.vozza@uniroma1.it (I.V.); livia.ottolenghi@uniroma1.it (L.O.); fabrizio.guerra@uniroma1.it (F.G.); 2Department of Oral Medicine and Radiology, Faculty of Dental Sciences, Sri Ramachandra Institute of Higher Education and Research, Chennai 600116, India; cvdivyambika@sriramachandra.edu.in; 3Medical Oncology Unit, Department of Radiological, Oncological and Anatomo-Pathological Sciences, Sapienza University, 00161 Rome, Italy; michela.roberto@uniroma1.it (M.R.); silverio.tomao@uniroma1.it (S.T.); 4Department of Interdisciplinary Dentistry, Pomeranian Medical University in Szczecin, 70111 Szczecin, Poland; maciej.jedlinski@pum.edu.pl; 5Department of Oral Surgery, Pomeranian Medical University in Szczecin, 70111 Szczecin, Poland; g.trybek@gmail.com

**Keywords:** imaging-based techniques, potentially malignant oral lesions, OPMD, diagnosis

## Abstract

Objectives: Oral potentially malignant disorders (OPMDs) are lesions that may undergo malignant transformation to oral cancer. The early diagnosis and surveillance of OPMDs reduce the morbidity and mortality of patients. Diagnostic techniques based on medical images analysis have been developed to diagnose clinical conditions. This systematic review and meta-analysis aimed to evaluate the efficacy of imaging-based techniques compared to the gold standard of histopathology to assess their ability to correctly identify the presence of OPMDs. Design: Literature searches of free text and MeSH terms were performed using MedLine (PubMed), Scopus, Google Scholar, and the Cochrane Library (from 2000 to 30 June 2020). The keywords used in the search strategy were: (“oral screening devices” or “autofluorescence” or “chemiluminescence” or “optical imaging” or “imaging technique”) and (“oral dysplasia” or “oral malignant lesions” or “oral precancerosis”). Results: The search strategy identified 1282 potential articles. After analyzing the results and applying the eligibility criteria, the remaining 43 papers were included in the qualitative synthesis, and 34 of these were included in the meta-analysis. Conclusions: None of the analyzed techniques based on assessing oral images can replace the biopsy. Further studies are needed to explore the role of techniques-based imaging analysis to identify an early noninvasive screening method.

## 1. Introduction

Oral potentially malignant disorders (OPMDs) are precursor lesions that may undergo malignant transformation to oral cancer. These lesions most commonly present clinically as white patches (leukoplakia). However, they may be red (erythroplakia) or red and white (erythroleukoplakia). Oral submucous fibrosis (OSMF) is another common OPMD seen predominantly in Southeast Asia and more commonly in the Indian subcontinent, presenting with severe burning sensation, blanching of the oral mucosa, and trismus due to fibrotic bands. OSMF is attributed to the use of areca nut, and it has been estimated that around 10–20% of the world population use areca nut in different formulations [1,2,3]. Currently, tobacco and areca nut usage in a variety of commercial preparations has led to an increase in the prevalence of OPMDs. The worldwide prevalence rate of OPMDs ranges from 1 to 5% [4,5].

Since India contributes to one-third of the global oral cancer burden, it is considered the world capital for oral cancer. Southern parts of India present the highest incidence rate of oral cancer among females both nationally and globally. The high financial burden for the patients undergoing oral cancer treatment leads to treatment breaks, thus adding further to the mortality rate [6]. The increased incidence of OPMDs and Oral Squamous Cell carcinoma (OSCC) in the Indian subcontinent is mainly attributed to the prevalence of tobacco usage among varied age groups. Apart from various forms of tobacco, chewing paan with areca nut contributes to oral malignancy, especially in the northeastern parts of India, leading to the high incidence of OSCC [7]. Global Adult Tobacco Survey (GATS) uses a standard and globally accepted protocol for monitoring adult smoking and smokeless tobacco use across countries, including India. According to the GATS 2 Fact Sheet, India 2016–2017, 42.4% of men, 14.2% of women, and 266.8 million of all adults currently use either smoking or smokeless forms of tobacco. One out of every eight young individuals 15 to 24 years old was using tobacco in any formulation.

Approximately 16–62% of OPMDs undergo a malignant transformation and eventually develop into oral squamous cell carcinoma (OSCC). OSCC is the most common oral cancer, representing more than 90% of all oral cancers. The five-year survival rates of OSCC patients decrease from 80 to 40% between diagnosis at an early or advanced stage [8]. Despite current advances in treatment, the 5-year survival rate for oral cancer has not increased substantially during the past several decades. The lack of effective early detection of high-risk OPMDs is one of the critical reasons for the poor prognosis of OSCC [9]. Treatment is more effective in patients with the early stage of disease; however, most patients present with advanced tumors for which treatment is less successful and may cause severe deficits in speech, swallowing, facial appearance, and quality of life [10].

The early diagnosis and surveillance of OPMDs reduce morbidity and mortality in OSCC patients. The early identification of dysplasia helps the clinician monitor the patients periodically and plays an important role in detecting oral carcinogenesis, improving the survival rate, and reducing disfigurement, loss of function, treatment duration, and the associated treatment expenditure, especially in developing countries worldwide [8].

Oral tissue biopsy is the gold standard for the diagnosis of OPMDs. However, this procedure is: (i) locally invasive; (ii) hardly acceptable by patients; (iii) time-consuming with no immediate results; (iv) primarily available in hospitals; and (v) not suitable for OPMDs long-term monitoring [11].

In this clinical scenario, diagnostic techniques based on medical images analysis have been developed using different approaches of artificial intelligence. They have been employed in different medical fields to detect, for instance, breast cancer in mammography, skin cancer in clinical skin screenings, diabetic retinopathy, in retinographies of periodontal bone loss on periapical and panoramic radiographs, and apical lesions and caries lesions on periapical radiographs. In oral pathology, several image-based techniques have been developed to detect OPMDs, the autofluorescence, high-resolution microendoscope (HRME), optical spectroscopy, narrow band imaging and vital staining colorants, among others [12,13,14,15,16].

These techniques are non-invasive, highly accepted and tolerated by patients, available in all clinical settings, non-operator-dependent, and repeatable [4,17,18,19,20,21,22,23].

Therefore, this systematic review and meta-analysis aimed to evaluate the efficacy of imaging-based techniques compared to the gold standard of histopathology in order to assess the ability of the imaging-based techniques to correctly identify the presence of OPMDs.

## 2. Materials and Methods

This systematic review was conducted according to the PRISMA (Preferred Reporting Items for Systematic Reviews and Meta-Analyses) statement and the guidelines from the Cochrane Handbook for Systematic Reviews of Interventions. The study protocol was registered after the screening stage (PROSPERO CRD42021230814).

### 2.1. Eligibility Criteria

The following inclusion criteria were applied to this meta-analysis: (a) randomized clinical trials (RCTs); (b) clinical trials; (c) cohort studies; (d) cross-sectional studies; (e) case-control studies; (f) pilot studies; (g) prospective and observational studies; (h) all considered participants were patients with suspicious oral lesions, a history of previously treated OSCC (Oral Squamous Cell Carcinoma) with no current evidence of cancer recurrence at least six months after cessation of treatment, or the presence of recently diagnosed, untreated OSCC or precancerous lesions; (i) the control intervention considered healthy volunteers without any oral abnormalities; (j) studies published in English, French, German, Spanish, Polish, Albanian, and Romanian. Broad inclusion criteria have been used to be as sensitive as possible. Exclusion criteria were as follows: (a) in vitro RCTs; (b) lack of effective statistical analysis; (c) abstract and author debates or editorials.

The outcomes to be assessed are listed as follows: sample size; oral lesions; presence or absence of biopsy; analyzed techniques for detection of mucosa alteration; strengths and weakness of each analyzed technique; measures: accuracy, sensitivity, specificity, positive predictive value, and negative predictive value.

### 2.2. Search Strategy and Study Selection

Literature searches of free text and MeSH terms were performed using PubMed, Scopus, Google Scholar, and the Cochrane Library (from 2000 to 30 June 2020). All searches were conducted using a combination of subject headings and free-text terms. The final search strategy was determined through several presearches. The keywords used in the search strategy were as follows: (“oral screening devices” or “autofluorescence” or “chemiluminescence” or “optical imaging” or “imaging technique”) and (“oral dysplasia” or “oral malignant lesions” or “oral precancerosis”).

The detailed search strategy used in PubMed, Scopus, Google Scholar, and the Cochrane Library are illustrated in Figure 1.

Reference lists of primary research reports were cross-checked to identify additional studies. Following the inclusion criteria, two authors (CA and AN) independently selected the literature by reading the titles and abstracts. The full text of each identified article was then read to determine whether it was suitable for inclusion. Disagreements were resolved through consensus or by a discussion with a third author (MM).

### 2.3. Data Collection

For each eligible study, data were independently extracted by two authors (CA and AN) and examined by the third author (MM). The data were compared through a created piloted spreadsheet, according to the Cochrane Collaboration guidelines. In case of missing data, MM contacted the corresponding author of the related research via e-mail and excluded those for which no reply was given.

### 2.4. Data Items

The following data items were recorded: study year, type and setting; age, size, and recruitment sample; case and control interventions; any pre-treatment and co-intervention; biopsy; analyzed technique; washout period in RCTs with crossover design; the follow-up, dropout and sample size at follow-up.

### 2.5. Quality Assessment

According to the PRISMA statements, the evaluation of the methodological quality indicates the strength of evidence provided by the study because methodological flaws can result in biases. For the randomized clinical trials, according to the Jadad scale, this procedure provides a total score that can range from 0 to 5, where 0 is a low-quality study, and 5 is the highest possible quality. A trial is considered to have a good quality when it receives a score of at least 3. For cross-sectional, case-control, and cohort studies, according to the Newcastle-Ottawa scale (NOS), the possible quality assessment score ranges from 0 to 9 points, with a high score indicating a good quality study.

### 2.6. Risk of Bias in Individual Studies

Selection bias (retained allocation concealment), performance and detection bias (blinding of participants and operators), attrition bias (patient dropout, washout period of crossover trials and missing values or participants, too short a duration of follow-up), and reporting bias (selective reporting, unclear eliminations, missing results) were recorded, evaluated, and allocated according to Cochrane guidelines [24].

### 2.7. Consistency Measures and Risk of Bias across Studies

Heterogeneity was assessed quantitatively using I2 170 -statistics and Cochran’s Q test [25]. The high percentage of variability comes from the heterogeneity of samples among studies. Funnel plot analyses were performed to assess small study effects or publication bias for analyses with two or more studies being present.

## 3. Results

### 3.1. Study Selection

The search strategy identified 1282 potential articles: 200 from PubMed, 18 from Scopus, 344 from Cochrane, and 720 from Google Scholar. After the removal of duplicates, 1045 articles were analyzed.

Subsequently, 937 papers were excluded because they did not meet the inclusion criteria. Of the remaining 108 papers, 65 were excluded because they were not relevant to the subject of the study. The remaining 43 papers were included in the qualitative synthesis, and 34 of these were included in the meta-analysis (Figure 1). Table 1 summarizes the characteristics of each of the 34 included studies. All the included papers reported odd ratio (OR) for the study’s relevant query data.

Meta-analysis was carried out for the sensitivity of techniques. Results were divided into five groups of used techniques: autofluorescence, HRME, optical spectroscopy, narrow band imaging, vital staining colorants. Meta-analysis was performed using a random-effect model. Effect sizes were calculated based on numbers of “Biopsy positive” and “Sample size positive (with biopsy positive)” true positives. The risk difference (RD) was taken as an effect size assuming zero risk of biopsy measurement. Heterogeneity was assessed quantitatively using I2 170 -statistics and Cochran’s Q-test [25]. Meta-analysis for the specificity of techniques was not carried out because false positives were only available for a few studies. Reported specificity and other characteristics were often calculated relative to techniques other than biopsy or a larger group, making results incomparable. The R statistical program (The R Foundation for Statistical Computing, Wirtschaftsuniversität Wien, Vienna, Austria) with compute.es and metaphor packages was used for the calculations [26,27].

**Table 1 ijerph-18-11775-t001:** Characteristics of included studies.

Author	Technique		Total Sample Size (Sites)	Sample Size Positive (with Biopsy Positive)	Biopsy (nr)	Biopsy Positive	Recall (Sensitivity)	Specificity	Precision (Positive Predictive Value)	Negative Predictive Value	Acc.	Confidence Interval	Confidence Interval (Sensitivity)	Confidence Interval (Specificity)	Confidence Interval PPV	Confidence Interval NPV
Lane 2006 [28]	Autofluorescence	Tissue fluorescence	50	43	50	44	98%	100%								
		Biopsy	50	44	50	44	100%	100%								
Mehrotra 2010 [29]		Vizilite	102	0	102	4	0%	75.5%	0%	94.8%			0–60.2%	66.7–82.8%	0–14.3%	89.9–99.9%
		Velscope	156	6	156	12	50%	39.9	6.4%	90.3%			21.1–78.9%	30.8–46.9%	2.4–13.4%	82.8–97.9%
Farah 2012 [30]		Velscope	118	80	118	112	63%	30%	19%	75%	55%					
		Biopsy	118	112	118	112	100%	100%	100%	100%						
Babiuch 2012 [31]		Velscope	50	12	50	12	100%	12.5%								
		Biopsy	50	12	50	12	100%	100%								
Rana 2012 [32]		Velscope	92	6	92	6	100%	74%					61–100%	67–82%		
		Biopsy	92	6	92	6	100%	100%								
Hanken 2013 [33]		Velscope	60	47	60	48	97.9%	41.7%					94–100%	14–70%		
		Biopsy	60	48	60	48	100%	100%								
Francisco 2014 [34]		Fluorescence spectroscopy	99	49	99	55	88.5%	93.8%								
		Biopsy	99	55	99	55	100%	100%								
Petruzzi 2014 (mild dysplasia as positive lesion) [35]		Autofluorescence	56	21	56	30	70.00%	57.69%	65.62%	62.50%	64.29%	2.6–53.0%				
		Toluidine Blue	56	24	56	30	80.00%	61.54%	70.59%	72.73%	71.43%	18.3–65.7%				
Petruzzi 2014 (mild dysplasia as negative lesion) [35]		Autofluorescence	56	13	56	17	76.47%	51.28%	40.62%	83.33%	58.93%	0.7–43.7%				
		Toluidine Blue	56	15	56	17	88.24%	51.28%	44.12%	90.91%	62.50%	11.1–50.5%				
Scheer 2016 [36]		Velscope	41	2	41	6	33.3%	88.6%	33.3%	88.6%						
		Biopsy	41	6	41	6	100%	100%	100%	100%						
Adil 2017 [37]		Velscope	90	31	90	34	76.6%	80.00%	97.5%	25.00%						
		Toluidine Blue	90	30	90	34	97.00%	67.00%	96.77%	66.66%						
Ganga 2017 [38]		Velscope	200	19	200	25	76.00%	66.29%	24.36%	95.08%			54.87–90.64%	58.76–73.24%	9.22–30.36%	90.52–97.51%
		Bi0psy	200	25	200	25	100%	100%	100%	100%						
Cânjău 2018 [39]		Velscope	18	16	18	17	94.44%	100%	100%	50%						
		Biopsy	18	17	18	17	100%	100%	100%	100%						
Chiang 2019 [40]		Autofluorescence	126	5	126	6	87.50%	72.73%	94.23%		85.07%					
		Biopsy	126	6	126	6	100%	100%	100%	100%						
Johnson 2019 [41]		Fluorescence	100	19	100	19	100%	80%								
		Biopsy	100	19	100	19	100%	100%								
Pierce 2012 [42]	HRME	autofluorescence imaging (AFI) and high-resolution microendoscope (HRME)	100	52	100	55	95%	98%								
		Biopsy	100	55	100	55	100%	100%								
Quang 2017 [43]		Autofluorescence imaging system (AFI) and a high-resolution microendoscope (HRME)	114	83	114	114	72.8%	100%								
		Biopsy	114	114	114	114	100%	100%								
Jo 2018 [44]		Endogenous Fluorescence Lifetime imaging	73	19	73	20	95.00%	86.00%		98.00%						
		biopsy	73	20	73	20	100%	100%		100%						
Yang 2018 [45]		HRME	56	51	56	56	91%	93%								
		Biopsy	56	56	56	56	100%	100%								
Yang 2020 [46]		Multimodal optical imaging	4	3	4	4	75%									
		Biopsy	4	4	4	4	100%									
Müller 2003 [47]	Optical Spectroscopy	Optical spectroscopy	91	32	91	44	74.00%	90.00%								
		Biopsy	91	44	91	44	100%	100%								
McGee 2009 [48]		Optical spectroscopy	87	7	87	14	53%	70%								
		Biopsy	87	14	87	14	100%	100%								
Schwarz 2009 [10]		Optical spectroscopy	154	52	154	63	82.00%	87.00%								
		Biopsy	154	63	154	63	100%	100%								
Sharwani 2006/1 [49]		Optical spectroscopy	71	30	71	33	90.3%	79.00%								
		Biopsy	71	33	71	33	100%	100%								
Sharwani 2006/2 [50]		Optical spectroscopy	25	8	25	11	72.00%	75.00%								
		Biopsy	25	11	25	11	100%	100%								
Jayanthi 2011 [51]		Optical spectroscopy	96	49	96	50	98.5%	96%	98.5%	96%						
		Biopsy	96	50	96	50										
Murdoch 2014 [52]		Optical spectroscopy	23	15	23	23	65.2%	62.5%								
		Biopsy	23	23	23	23										
Yang 2012 [53]	Narrow band imaging	Narrow band imaging	74	47	74	55	84.62%	94.56%	74.32%	97.06%	93.00%		75.84–93.39%	92.18–96.94%	64.37–84.28%	95.26–98.85%
		Biopsy	74	55	74	55	100%	100%	100%	100%	100%					
Upadhyay 2019 [54]		Narrow band imaging	32	29	32	31	93.93%	80%	31	66.66%						
		Biopsy	32	31	32	31	100%	100%	100%	100%						
Nagaraju 2010 [55]	Vital staining colorants	Toluidine blue	60	30	60	30	100%%	60.00%	100%	43.00%						
		Biopsy	60	30	60	30	100%	100%	100%	100%						
Güneri 2011 [56]		Toluidine blue	42	6	42	15	92.3%	43.3%	41.4%	92.9%						
		Biopsy	42	16	42	15	100%	100%	100%	100%						
Prajeesh 2019 [57]		Toluidine blue	183	134	183	134	96.4%		91.8%	86.5%						
		Biopsy	183	139	183	139	100%		100%	100%						
Epstein 2003 [58]		Tolonium	96	29	96	30	96.7%		36.4%							
		Biopsy	96	30	96	30	100%		100%							
Bhalang 2008 [59]		Acetic acid	55	27	55	33	83.33%	84.21%	90.91%	72.73%	83.64%					
		Biopsy	55	33	55	33	100%	100%	100%	100%	100%					
Qaiser 2020 [60]		Flureiscein dye	100	38	100	40	95%									
		Biopsy	100	40	100	40	100%									

### 3.2. Study Characteristics

Included studies (Table 1) were published between 2006 and 2020, and they were focused on the Autofluorescence technique (n:14) [28,29,30,31,32,33,34,35,36,37,38,39,40,41], HRME (n:5) [42,43,44,45,46], Optical spectroscopy (n:7) [10,47,48,49,50,51,52], NBI (n:2) [53,54], and Vital Stain Colorants (n:6) [55,56,57,58,59,60]. In total, 34 studies were performed in adults (18 years of age or older). Sample sizes ranged between 4 and 200 participant sites (mean: 82), for an overall sample size of 2792. Each technique was compared to the golden standard: tissue histology that always guarantees 100% diagnosis.

#### 3.2.1. Autofluorescence

Autofluorescence is one potential technique that may be used to facilitate the visualization of OPMD and oral cancer. Autofluorescence works on the principle that certain biofluorophores present within the tissue become fluorescent on excitation with a suitable wavelength (400–460 nm) light source. The diseased tissues tend to appear darker since they lose fluorescence, which is attributed to the disruption in the distribution of the biofluorophores [61]. In the present study, the image-based techniques based on autofluorescence were: Vizilite and VELScope. Wide-field imaging devices, such as the VELScope, can survey large mucosal surface areas to detect regions with loss of autofluorescence, which is suspicious for dysplasia.

#### 3.2.2. High-Resolution Microendoscopy (HRME)

The high-resolution microendoscope (HRME) uses a coherent fiber bundle to obtain high-resolution fluorescence images of the tissue in contact with the distal tip of the device without the need for complex mechanical scanning systems and associated control electronics. The system uses a low-cost light-emitting diode to provide illumination and a consumer-grade charge-coupled device camera to capture high-resolution digital images on a laptop computer [62,63]. HRME is usually used together with autofluorescence imaging (AFI) because HRME devices can complement these types of wide-field imaging systems by providing high-resolution image data at specific lesions first identified by loss of fluorescence [64].

#### 3.2.3. Optical Spectroscopy

Optical spectroscopy can be used to detect changes in oral mucosa during carcinogenesis in the oral cavity, based on the fact that increased nuclear size and nuclear to cytoplasmic ratio, increased microvascularization, degradation of stromal collagen, and alterations in the concentration of mitochondrial fluorophores such as reduced nicotinamide adenine dinucleotide (NADH) and flavin adenine dinucleotide (FAD) affect the optical scattering, absorption, and autofluorescence characteristics within the tissue. Spectroscopic measurements are usually performed in a darkened room to minimize the effects of ambient light. The optical spectroscopy autofluorescence spectra are characterized by 12 excitation wavelengths ranging from 300–470 nanometers (nm) and a diffuse reflectance spectrum under white light illumination that can be collected through each of four probe channels with different depth responses, for a total of 52 spectra collected in each 90 s measurement. The shallow channel has a depth response weighted toward the epithelial tissue layer; the medium channel interrogates both epithelium and shallow stroma, and the two deep channels collect signals primarily from the stroma [10].

#### 3.2.4. Narrow Banding Imaging

Narrow band imaging (NBI) is an endoscopic technique based on the use of special optical filters that narrow the light bandwidth to enhance the visualization of the mucosa surface and microvasculature [65]. NBI associated with a magnification zoom facility is useful for the accurate diagnosis of early cancers due to the contrast observation of vascular architecture, particularly the intrapapillary capillary loops (IPCL) [65]. In neoplastic lesions, the IPCL are modified by dilation, meandering, and caliber irregularities that can be differentiated from normal mucosa. Morphological changes to the IPCL are useful for diagnosing early cancers and determining the depth of invasion [66] and the margin of resection [67,68].

#### 3.2.5. Vital Stain Colorants

The vital colorants have been used as diagnostic aids since the early 1980s. The changes that characterize dysplastic and malignant cells (disordered arrangement, loose connections, more nucleic acid than normal cells) allow the colorants to enter the extracellular space and bind the intracellular nucleic acid making the tissue distinguished [69]. This study report results mainly on toluidine blue, tolonium, acetic acid, and fluorescein dye. Toluidine blue stain could serve as an adjunct during clinical examination for recognition of suspicious lesions which could undergo malignant transformation [70]. Vital staining techniques also aid in choosing the appropriate site for biopsy of both large lesions and when there is multiple site involvement and has been suggested to be an effective screening tool, especially in high-risk cases [71]. Although some of the potential disadvantages include the interpretation of faintly positive cases, adequate training and colour guide may help overcome the disadvantages, thereby improving the sensitivity and specificity [72].

### 3.3. Quality Assessment

According to the Jadad scale for RCT, the authors evaluated the quality of one clinical trial [33] included in the qualitative synthesis, based on five questions that analyze the randomization process, the experimental blinding, and the dropout rate, i.e., the patients lost to follow-up. In evaluating the quality of RCTs, the total score of this study was 3, indicating a good quality study (Table 2).

According to the Newcastle-Ottawa scale (NOS) on cross-sectional (n:24) [28,29,31,35,37,39,40,42,43,44,45,46,48,49,50,54,55,56,59,60,73,74,75,76]; case-control (n:10) [10,32,47,51,52,53,77,78,79,80] and cohort studies (n:7) [30,34,36,38,41,57,58] the authors evaluated the qualities of all included studies based on object selection, comparability, and exposure. A star was described as an appropriate entry, with each star representing one point. The possible quality assessment score ranged from zero to nine points, with a high score indicating a good quality study. In evaluating the quality of cross-sectional studies, the total scores of two studies were lower or equal to five, indicating low-quality studies, while the total scores of the other twenty-two were six or higher, indicating medium or high-quality studies (Table 3). For the case-control studies, the total scores of three studies were lower or equal to five, indicating low-quality studies, while the total scores of the other seven were six or higher, indicating medium or high-quality studies (Table 4). While the quality of cohort studies, the total score of all included studies, except for one, was greater than or equal to 6, indicating high-quality studies (Table 5).

### 3.4. Effect Size

A comparison of studies is shown on the forest plot (Figure 2). Using of techniques has a small negative effect size of 81 in all five groups: from—0.10 in narrow band imaging and Vital staining colorants groups to—0.20 in the optical spectroscopy group. There are no significant differences in RD between groups (minimum p-value equals 0.28 in Welch’s two sample *t*-tests).

### 3.5. Consistency and Publication Bias

Heterogeneity is significant at *p* ≤ 0.001 level in all studies and all but a very small NBI group. From 77.6% of the variability in the HRME group to 87.6% in the vital staining colorants group come from heterogeneity. Consistency in the NBI group should be taken cautiously because the Q-based test is known to be poor at detecting true heterogeneity when the number of studies is small [81].

Funnel plot (Figure 3) shape suggests publication bias—results with larger risk difference having more chances to be published even if standard error is significant (i.e., number of positives is small).

### 3.6. Summarizing Findings

Calculations showed:-Using of techniques has a small negative effect size in sensitivity compared to biopsy. There are no significant differences in risk difference between techniques.-Results are inconsistent both in the whole and in technique groups.-Evidence of publication bias was also detected by analyzing funnel plot.-Results reporting is not homogeneous across studies, which makes it challenging to carry out a reliable comparison of measures like specificity or positive/negative predictive values.

## 4. Discussion

The current systematic review and meta-analysis reported on the sensitivity of in vivo imaging-based techniques for early diagnosis of potentially malignant oral lesions (OPMDs) versus biopsy, which is considered the gold standard. The results showed that using of techniques has a small negative effect size in all five groups: from—0.10 in NBI and vital staining colorants groups to—0.20 in the optical spectroscopy group. Studies with a lower sampling number showed worse results. With larger sample sizes, sensitivity also increased. The overall meta-analysis, however, reported a small negative effects size (−0.14; −0.18; −0.10; 95%CI) of these techniques when compared to histopathology.

Oral screening is the first stage of early diagnosis: it can avoid delayed referrals and therefore reduce the mortality of SCC. It has been reported that SCC may develop from an OPMD, and its diagnosis is an important preventive step with significant consequences on patient survival rate and quality of life. The major limitation of visual inspection is the difficulty differentiating between benign, high-risk lesions and other mucosal diseases and conditions in the oral cavity [82,83].

Tissue biopsy is still considered as a gold standard for SCC diagnosis, however, it is invasive, time-consuming, painful, operator dependent, not easily acceptable by patients, especially by those with prior history of cancer, and available mainly in the hospital settings despite the need for oral screening to be widely performed in the private dental practices daily [4]. In fact, most SCCs tend to arise during the first two years after the detection of an OPMD, but several follow-up studies show that the risk may last up to 10–15 years. Therefore, managing these conditions may require regular and long-term follow-up and repeated diagnostic procedures by an experienced oral health professional. Erythroplakia, erythroleukoplakia, proliferative verrucous leukoplakia, and oral submucous fibrosis present with high malignant transformation risk that is lower in leukoplakia and oral lichen planus [4,84].

Diagnostic imaging-based techniques claim for higher acceptability by patients, low invasiveness, immediate results, repeatable assessments, and are suitable for regular follow-up and availability in the private practice setting and to be non-operator dependent. Some of these techniques are already commercially available, such as Vizilite and VelScope, and their accessibility also depends on the possibility to be purchased from health facilities, even by smaller ones. Moreover, these techniques reflect the current trend in healthcare toward artificial intelligence that can successfully transform medical images analysis in pathology, radiology and other fields of medical imaging [85,86,87]. A review published in 2018 by Yang et al. [45] on noninvasive diagnostic adjuncts for OPMDs evaluation showed that these techniques presented poor diagnostic accuracy due to high false-positive results [45]. Therefore, the current systematic review included studies published in the last fifteen years (2006–2020) to evaluate these techniques’ further development, clinical applications and possible improvement/amelioration.

Starting with the best performance, NBI and Vital Staining colorants were characterized by a −0.10 RD and (−0.18, −0.03) and (−0.18, −0.02) 95%CI, respectively. The group of NBI accounted for two studies published in 2019 and 2012 [53,54]. In the study by Yang et al. [54], the total sample size was 74 sites with biopsy and 74 with NBI. The sensitivity and specificity were 84.62% and 94.56%, respectively [53]. The sample size in the study by Upadhyay was 32 sites with biopsy and 32 with NBI, and the sensitivity and specificity resulted in 93.93% and 80% [54]. Our data agree with a recent meta-analysis aiming to describe the validity of NBI in the assessment of suspicious oral lesions demonstrated a specificity and sensitivity of 75.7% with 95% CI 65.1–83.9% and 91.5% with 95% CI 81.8–96.3%, respectively [88]. The meta-analysis authors concluded that NBI could play a decisive role in a surveillance setting for low-risk lesions or lesions for which multiple biopsies may not be practical. Within higher-powered prospective studies with the establishment of precise clinical recommendations, the full potential of NBI both in the screening and tertiary referral setting could then be realized.

The vital staining colorants group evaluated the sensitivity in diagnosing OPMDs in 8 studies, published between 2008 and 2020, for a total sample size of 536 sites evaluated twice (Vsc vs. biopsy). Only data on sensitivity was reported in all the trials, while specificity only by three on them. The lowest sensitivity was shown in the study by Bhalang et al. [59] that evaluated the application of acetic acid in a group of 30 participants. This study scored seven in the quality assessment with the Newcastle-Ottawa scale adapted for cross-sectional studies [59]. The authors concluded that the results were promising, and the acetic acid was conveniently obtainable at any market despite the other colorants such as toluidine blue and recommended its use especially for oral screening in rural communities. Toluidine blue was the most frequently used colorant, and its best sensitivity performance was 92.3% in the study by Güneri et al. [56] aiming to evaluate toluidine blue, oral brush cytology, and biopsy in a group of 35 subjects and 43 oral lesions [56]. The results showed that Toluidine blue is a noninvasive method that offers real-time clinical information that may aid in completing a biopsy, biopsy site selection, and referral, in contrast to brush cytology, which requires specimen collection and a laboratory procedure. The absolute result was obtained by Fluorescein dye in the study conducted in 2020 by Qaiser et al. [60], with 95% of sensitivity. Fluorescein has been tested for the very first time for oral diagnosis of OPMDs, while it has been previously used for cancer diagnosis in the stomach, colon, breast, and brain [60]. The study was conducted at a tertiary care dental center and included 100 individuals. The dye is of organic origin, yellow, hypoallergenic, and nontoxic for topical use and produces an intense green-fluorescence in slightly acid to alkaline solutions. The study concluded that Fluorescein is an affordable and accessible technique, and it may find a promising role in community-based screening programs for oral cancer, especially in low- and middle-income regions, due to it being a safe and cost-effective measure [60,89].

It is noticeable that OSCC is the most commonly reported malignancy in Southeast Asia for the highest cancer mortality, according to the GLOBOCAN 2018 [90]. Recent global estimates have revealed an annual incidence of 246,420 males and 108,444 females being registered worldwide.

The highest incidence rates for oral cancer are found in Southeast Asia, including the Indian subcontinent [90]. Currently, a distinct imbalance exists with regard to the oral cancer burden between the developed and the developing countries. This imbalance needs to be corrected [90,91], and it partly accounts for the design of the included studies involving the research from Southeast Asia and the Indian subcontinent.

The autofluorescence subgroup was the most numerous, with a total of 15 included studies being published between 2006 and 2019. The reported risk difference was −0.16 (95% CI −0.24–0.08). Vizilite, VELscope^®^, Autofluorescence, fluorescence, and fluorescence spectroscopy were included in this subgroup. VELscope^®^ was analyzed by nine studies, with a sensitivity ranging from 50% to 100%. In the study by Babiuch et al. VELscope^®^ showed a sensitivity and specificity for the detection of a dysplastic and cancer lesion of 100% and 12.5%, respectively, demonstrating that VELscope^®^ is useful in the detection of oral lesions, but not able to differentiate high risk and low-risk lesions [31]. The study by Rana et al. [32] showed excellent test results of 100% for the specificity of the device, but 74% for the specificity, explained by the fact that the study population consisted of patients with different histologic diagnoses and not only by high-risk patients with a former oral cancer diagnosis. The conclusions clearly underlined that VELscope^®^ cannot replace biopsy but could help detect the right location for the procedure [32]. Then, although many widely recognized organizations as FDA and WHO approved VELscope^®^ as a diagnostic tool for premalignant alternations and oral carcinoma, many researchers remain wary by stating that autofluorescence detecting devices can only act as an adjuvant and cannot be used as a confirmatory test in the diagnosis of oral cancer [92]. Moreover, as the study by Rana et al. [32] demonstrated, the device used by unskilled examiners could lead to high false-positive results due to misinterpretation of the test. In addition, Amirchaghmaghi et al. [93] stated that this device should not be used at a screening stage, as the high probability of false-positive results may cause a high unnecessary referral rate and unnecessary biopsies [93].

ViziLite is based on chemiluminescence; it was first used in the cervix to detect dysplasia and was recently adapted for examination of the oral mucosa. The device is easy to use, does not require consumable reagents, and provides real-time results. On the other hand, VizLite needs a dark environment, has high recurrent costs especially problematic for low-income regions. It leaves no permanent record unless photographed, carries low specificity for dysplasia, results in high referral rate and over-treatment, and is unable to detect some red lesions. The study by Mehrota et al. [29] reported a specificity of 75.5%, and the authors concluded that ViziLite does not have the ability to accurately classify OPMDs by discriminating between high-risk and low-risk lesions and therefore should be used with caution. Moreover, since it is not able to detect some red lesions, it is reasonable to suggest that ViziLite assessments be interpreted in conjunction with the clinical findings [29]. The discussion on the risk of bias among the studies in the Autofluorescence aligns well with the recent systematic review of the bias found within studies evaluating the autofluorescence to detect OPMDs and OSCC, conducted by Tiwari et al. [92]. Among others, unclear descriptions of inclusion and exclusion criteria and small sample size were listed [94].

The high-resolution microscope (HRME) subgroup analyzed the data from five studies published from 2012 to 2020. The RD was −0.12 with 95%CI −0.22, −0.02. Two studies were carried out by Yang et al. in 2018 and have demonstrated that the combined approach of autofluorescence (AFI) and high-resolution microscope (HRME), defined as multimodal imaging, showed higher accuracy than either modality alone, with 91% sensitivity in diagnosing OPMDs and OSCC [45,46]. Autofluorescence is noninvasive, macroscopic imaging of mucosa, showing high sensitivity for the detection of dysplasia and cancer, based on loss of fluorescence; however, benign lesions frequently gave a similar appearance to the dysplastic lesions by showing loss of fluorescence due to stroma; inflammation, thus leading to false positives [95]. HRME is a flexible fibre-optic fluorescence microscope that can image nuclear features of dysplasia such as nuclear to cytoplasm (NC) ratio and nuclear crowding using the topical contrast agent proflavine. Although HRME images have improved specificity even in the presence of stromal inflammation, it has a <1 mm field of view, which poses difficulty in assessing entire lesions [42,43,62]. Hence it would be more appropriate for multimodal imaging where clinicians could exploit the different modalities of imaging based on its indications and advantages.

In 2018, Yang et al. [45] concluded that AFI and HRME could be used in an integrated manner for large suspicious lesions such that AFI first aids in the detection of the high-risk site within a large lesion followed by HRME imaging so that the entire lesion can be assessed for selecting the biopsy site [45]. This multimodal imaging can be effectively employed for community screening, thereby facilitating early detection and the reduction of oral cancer burden [45]. In 2020 Yang et al. carried out the subsequent clinical study and demonstrated that multimodal optical imaging identified more cases of high-grade dysplasia than clinical evaluation alone, but also indicated that a negative result in a high-risk population may not be a sufficient justification to avoid a clinically indicated biopsy [46]. The studies by Yang et al. reflected the status of multimodal optical imaging as an emerging technology that requires additional fine-tuning, further elucidation of its role in patient care, and randomized studies with larger sample sizes. More broadly, the optimal role of the MMIS has yet to be determined [46].

The optical spectroscopy subgroup in the meta-analysis of data shows RD—0.20 and 95% CI −0.30–0.09. The included studies were published between 2003 and 2014, with sensitivity ranging from 53% [50] to 98.5% [53]. Jayanthi et al., despite the good results, stated that large, prospective clinical studies are needed to reliably evaluate the efficacy of spectroscopy for these applications [51].

The diagnosis of oral cancer requires procedures with proven sensitivity and specificity, which are operator-independent and can be repeated in cases where high-risk patients require a long follow-up period. Data reported in this review are based on different techniques with high heterogeneity in study design, inclusion criteria, staging of disease and grading of differentiation (from 77.6% of the variability in the HRME group to 87.6% in vital staining colorants group come from heterogeneity). Therefore, prospective clinical trials reporting more homogeneous characteristics are needed. Moreover, diagnostic procedures must be adapted to national health systems, especially in those socio-political and economic macro areas characterized by fragility and difficulties in organizing and training highly qualified personnel. Thus, supporting research in underdeveloped countries can provide an opportunity to identify a technique with good clinical outcomes and low costs.

## 5. Limitations

The limitations are mainly due to the variable number of studies included for each analyzed technique, ranging from a minimum of 2 (NBI) to a maximum of 14 studies (Autofluorescence). In addition, heterogeneity was significant at the *p* ≤ 0.001 level in all included studies, while the funnel plot showed publication bias. Moreover, the high percentage of variation resulted from the heterogeneity of the samples among the studies.

## 6. Conclusions

In conclusion, based on this systematic review and meta-analysis, none of the here analyzed techniques based on assessing oral images can replace the biopsy that remains the gold standard in the diagnosis of OPMDs and OSCC. However, according to the promising results obtained with NBI studies, further research is needed to explore the role of other techniques based on AI and imaging analysis to identify an early noninvasive screening method.

## Figures and Tables

**Figure 1 ijerph-18-11775-f001:**
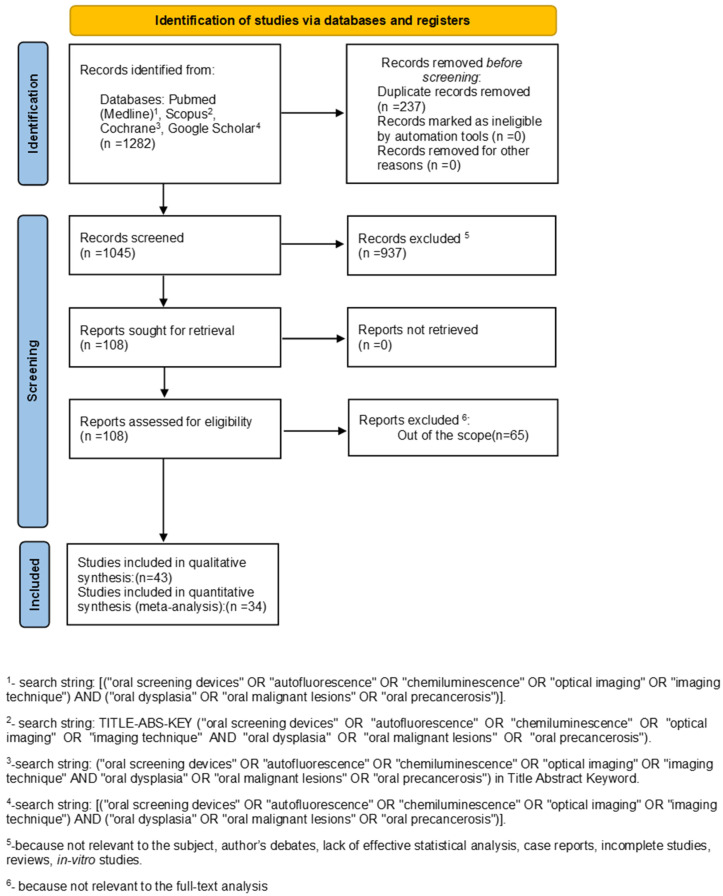
The flowchart of the search.

**Figure 2 ijerph-18-11775-f002:**
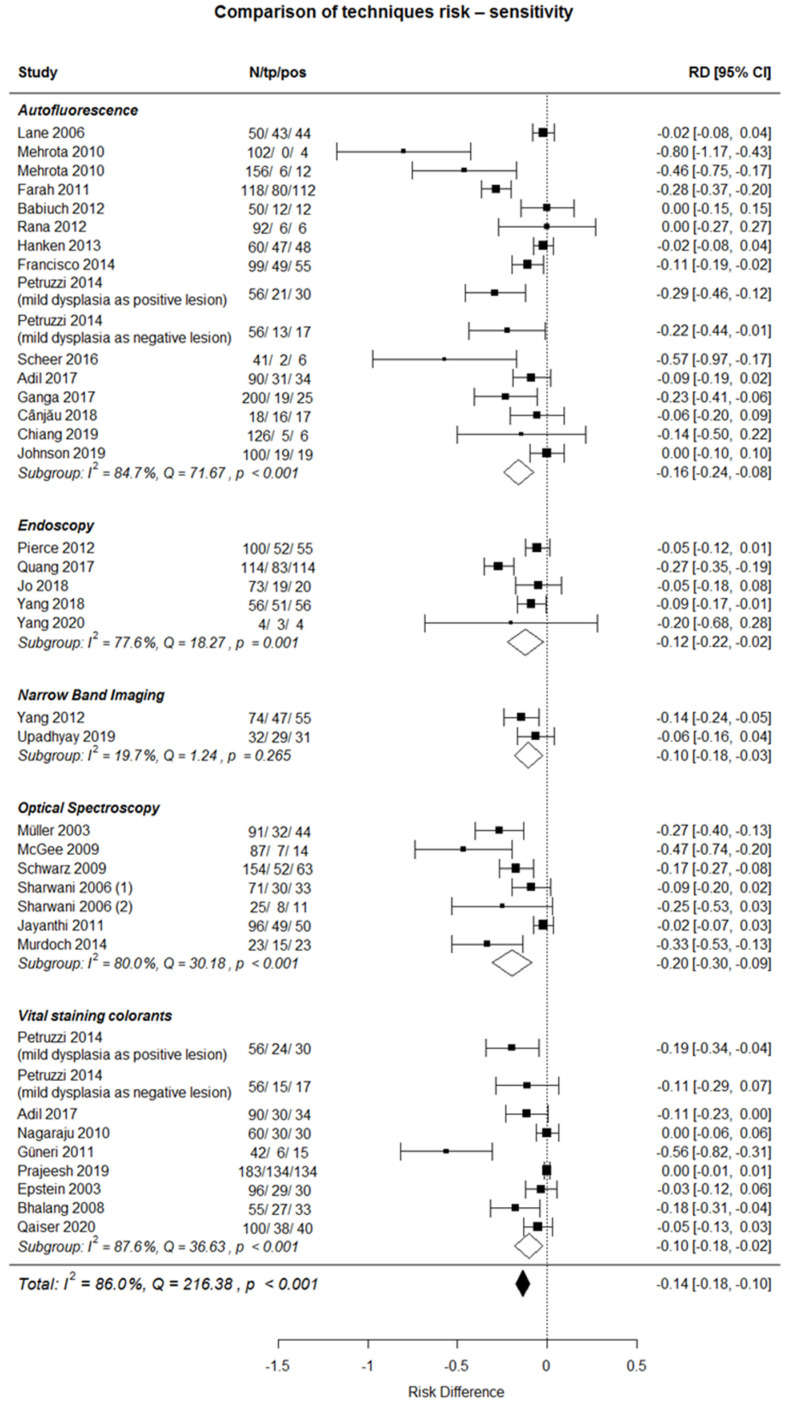
Comparison of studies.

**Figure 3 ijerph-18-11775-f003:**
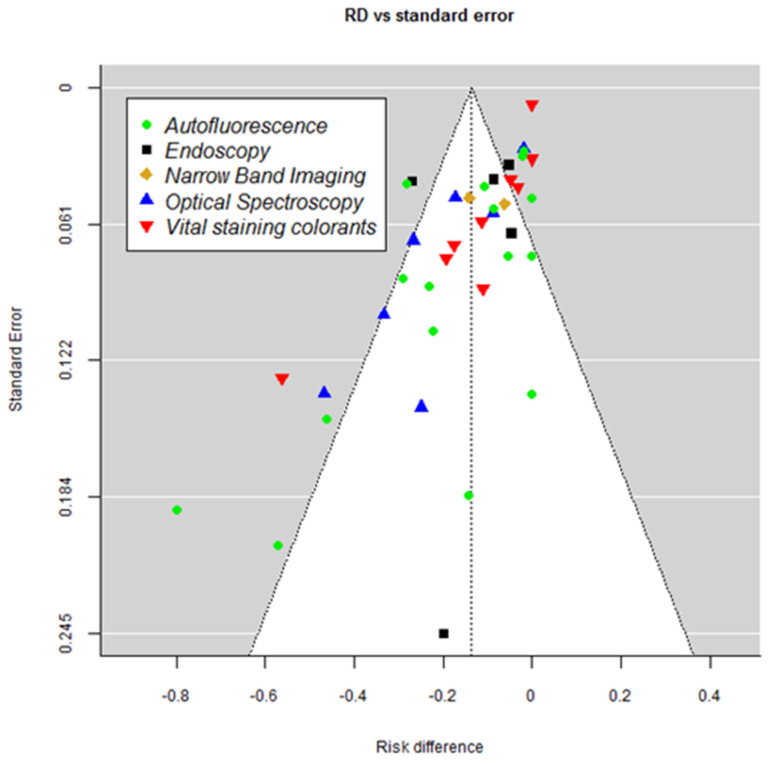
Funnel plot.

**Table 2 ijerph-18-11775-t002:** Jadad Scale for randomized control trials.

Jadad Scale for Reporting Randomized Controlled Trials	
	Hanken [33]
(1) Is the study described as randomized?	1
(2) Is the study described as double-blind?	0
(3) Is there a description of withdrawals and dropouts?	0
(4) The method of randomization is appropriate?	1
(5) The method of blinding is appropriate?	1
Total score=	3

**Table 3 ijerph-18-11775-t003:** Newcastle-Ottawa Scale adapted for cross-sectional studies.

Newcastle-Ottawa Scale Adapted for Cross-Sectional Studies																									
		Ambele 2020 [73]	Cânjău 2018 [39]	Mehrotra 2010 [29]	Adil 2017 [37]	Babiuch 2012 [31]	Ikeda 2020 [74]	Poh 2007 [75]	Lane 2006 [28]	Chiang 2019 [40]	Yang 2018 [45]	Pierce 2012 [42]	Yang 2020 [46]	Jo 2018 [44]	McGee 2009 [48]	Sharwani 2006 [49]	Sharwani 2006 [50]	Upadhyay 2019 [54]	Petruzzi 2014 [35]	Bhalang 2008 [59]	Güneri 2011 [56]	Nagaraju 2010 [55]	Fakurnejad 2019 [76]	Qaiser 2020 [60]	Quang 2017 [43]
**Selection:** (Maximum 5 stars)	(1) Representativeness of the sample			*	*		*		*	*		*	*	*	*	*									*
	(2) Sample size			*			*		*	*			*		*	*		*			*				*
	(3) Non-respondents	*	*	*	*	*	*	*	*	*	*	*	*	*	*		*	*	*	*			*		*
	(4) Ascertainment of the exposure (risk factor)	**	**	**	**	**	**	**	**	**	**	**	**	**	**	**	**	**	**	**	**	**	**	**	**
**Comparability:** (Maximum 2 stars)	(5) The subjects in different outcome groups are comparable, based on the study design or analysis. Confounding factors are controlled.	*	*	*	*	*	*	*	*	*	*	*	*	*	*	*	*	*	*	*	*	*	*	*	*
**Outcome:** (Maximum 3 stars)	(6) Assessment of the outcome	**	**	**	**	**	**	**	**	**	**	**	**	**	**	**	**	**	**	**	**	**	**	**	**
	(7) Statistical test				*		*			*						*		*	*	*	*		*		*
	Total score=	6	6	8	8	6	9	6	8	9	6	7	8	7	8	8	6	8	7	7	6	5	7	5	9

* The tool is available or described; ** Validated measurement tool.

**Table 4 ijerph-18-11775-t004:** Newcastle-Ottawa quality assessment scale case-control studies.

Newcastle-Ottawa Quality Assessment Scale Case-Control Studies											
		Poh 2016 [77]	Rana 2012 [32]	Huff 2009 [78]	Schwarz 2009 [10]	Jayanthi 2011 [51]	Murdoch 2014 [52]	Mallia 2010 [79]	Müller 2003 [47]	Yang 2012 [53]	Vijayavel 2013 [80]
**Selection:** (Maximum 4 stars)	(1) Is the case definition adequate?	*	*	*	*	*	*	*	*	*	*
	(2) Representativeness of the cases	*	*	*	*	*	*			*	*
	(3) Selection of Controls				*	*	*	*	*		*
	(4) Definition of Controls				*	*	*	*	*		*
**Comparability:** (Maximum 2 stars)	(5) Comparability of cases and controls based on the design or analysis	*	*	*	*	*	*	*	*	*	*
**Outcome:** (Maximum 3 stars)	(6) Ascertainment of exposure	*	*	*	*	*	*	*	*		*
	(7) Same method of ascertainment for cases and controls			*	*	*	*	*	*		*
	(8) Non-Response rate	*	*	*						*	*
	Total score=	5	5	6	7	7	7	6	6	4	8

* The answer is “yes” or it is described in the study.

**Table 5 ijerph-18-11775-t005:** Newcastle-Ottawa quality assessment scale cohort studies.

		Scheer 2016 [36]	Farah 2012 [30]	Ganga 2017 [38]	Francisco 2014 [34]	Prajeesh 2019 [57]	Epstein 2003 [58]	Johnson 2019 [41]
**Selection:** (Maximum 4 stars)	(1) Representativeness of the exposed cohort	*	*	*	*	*		*
	(2) Selection of the non-exposed cohort	*	*	*		*	*	
	(3) Ascertainment of exposure	*	*	*	*	*	*	*
	(4) Demonstration that outcome of interest was not present at the start of the study							
**Comparability:** (Maximum 2 stars)	(5) Comparability of cohorts based on the design or analysis	*	*	*	*	*	*	*
**Outcome:** (Maximum 3 stars)	(6) Assessment of outcome	*	*	*	*	*	*	*
	(7) Was follow-up long enough for outcomes to occur	*	*	*			*	*
	(8) Adequacy of follow up of cohorts	*	*	*	*	*	*	*
	Total score=	7	7	7	5	6	6	6

* Described or available in the study.

## Data Availability

The results of the search are available under URL.

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
