# Peer review of "In Vivo Imaging-Based Techniques for Early Diagnosis of Oral Potentially Malignant Disorders—Systematic Review and Meta-Analysis"

_ijerph, 2021, doi:10.3390/ijerph182211775_

Round 1
Reviewer 1 Report
The aim of this systematic review and meta-analysis were in vivo imaging-based techniques for early diagnosis of potentially malignant oral lesions.
Comments:
Introduction - the more detailed information about the imaging-based techniques for early diagnosis of potentially malignant oral lesions is needed. The search protocol, inclusion and exclusion criteria should be described in more detail.
In Discussion the authors should explain the limitations of the study and provide explanation for discrepancies.
Author Response
"Please see the attachment."

Reviewer 2 Report
This paper performed a systematic review and meta-analysis of currently used in vivo imaging techniques capable of early diagnosis of OPMD to reduce morbidity and mortality in OSCC patients. The experimental design and analysis have been done well, but it seems that a few things need to be corrected/improved as shown below.
- The fact that the highest incidence of oral cancer occurs in Southeast Asia, including the Indian subcontinent, does not seem to have much relevance to the motivation or experimental design of this study. Therefore, I don't think the author needs to re-emphasize this in the discussion.
- On page 7, lines 4-5, “Endoscopy” is mentioned twice. One of them should be replaced by "optical spectroscopy".
- In this paper, endoscopy was used only for HRME, but NBI (narrow band imaging) is also one of the endoscopic techniques. Therefore, it is recommended to refine the terminology to be specific.
- Since the number of analysis papers of NBI is too small (n=2), it is difficult to compare the sensitivity with other technologies.
- Vital stain colorants is a visualization technique through staining, not an "optical imaging technique". It is treated as separate from the previous four optical imaging methods.
- In Figure 3, autofluorescence and NBI have similar colors, making it difficult to distinguish them.
- Some of the parts mentioned in the introduction are repeated unnecessarily in the discussion.
Author Response
"Please see the attachment."

Reviewer 3 Report
The present paper deals with a very interesting topic, in a systematic and scientifically valid method.
An English language review is required by a native speaker expert.
Minor revisions are required.
- “The keywords used in the search strategy were as follows: [("oral screening devices" OR "autofluorescence" OR "chemiluminescence" OR "optical imaging" OR "imaging technique") AND ("oral dysplasia" OR "oral malignant lesions" OR "oral precancerosis")]. “ The queries used were the same for each database used? Please define if the mentioned query was used for each database with the same combination of parentheses, words, and Boolean operators. In case the same query has not been used for each database, then indicate the exact query in each of them.
- Using LILACS could improve the review as e.g. papers in Spanish could be identified (Abstracts are available in English). Please justify why no other databases other than those mentioned in the manuscript were used.
- Please define the potential limitations of this study using a dedicated paragraph.
Author Response
"Please see the attachment."

Round 2
Reviewer 2 Report
I appreciate the efforts of the authors for corrections and improvements. But still below two things need improvement.
- Since endoscopy is a broad technology including NBI, the terminology in Table 1 should be changed. It is recommended to use HRME instead of endoscope.
- In Figure 3, the lilac color looks very similar to the red on my monitor. There must be a better way to express it. It is recommended to change the symbol shape as well as the color to something other than circle.
Author Response
"Please see the attachment."
